# Pathobiological and Genomic Characterization of a Cold-Adapted Infectious Bronchitis Virus (BP-caKII)

**DOI:** 10.3390/v10110652

**Published:** 2018-11-19

**Authors:** Seung-Min Hong, Se-Hee An, Chung-Young Lee, Chang-Seon Song, Kang-Seuk Choi, Jae-Hong Kim, Hyuk-Joon Kwon

**Affiliations:** 1Laboratory of Avian Diseases, College of Veterinary Medicine, Seoul National University, Seoul 08826, Korea; topkin@snu.ac.kr (S.-M.H.); eepdl1201@snu.ac.kr (S.-H.A.); bluespell@snu.ac.kr (C.-Y.L.); kimhong@snu.ac.kr (J.-H.K.); 2Laboratory of Avian Diseases, College of Veterinary Medicine, Konkuk University, Seoul 05029, Korea; songcs@konkuk.ac.kr; 3Avian Disease Division, animal and Plant Quarantine Agency, Gimcheon, Gyeongsangbuk-Do 39660, Korea; kchoi0608@korea.kr; 4Reseach Institute for Veterinary Science, College of Veterinary Medicine, BK21 for Veterinary Science, Seoul 08826, Korea; 5Department of Farm Animal Medicine, College of Veterinary Medicine, Seoul National University, Seoul 08826, Korea; 6Farm Animal Clinical Training and Research Center, Seoul National University, Seoul 08826, Korea

**Keywords:** infectious bronchitis virus, cold adaptation, comparative genomics, premature reproductive tract pathogenicity model, persistent infection

## Abstract

We established a cold-adapted infectious bronchitis virus (BP-caKII) by passaging a field virus through specific pathogen-free embryonated eggs 20 times at 32 °C. We characterized its growth kinetics and pathogenicity in embryonated eggs, and its tropism and persistence in different tissues from chickens; then, we evaluated pathogenicity by using a new premature reproductive tract pathogenicity model. Furthermore, we determined the complete genomic sequence of BP-caKII to understand the genetic changes related to cold adaptation. According to our results, BP-caKII clustered with the KII genotype viruses K2 and KM91, and showed less pathogenicity than K2, a live attenuated vaccine strain. BP-caKII showed delayed viremia, resulting in its delayed dissemination to the kidneys and cecal tonsils compared to K2 and KM91, the latter of which is a pathogenic field strain. A comparative genomics study revealed similar nucleotide sequences between BP-caKII, K2 and KM91 but clearly showed different mutations among them. BP-caKII shared several mutations with K2 (nsp13, 14, 15 and 16) following embryo adaptation but acquired multiple additional mutations in nonstructural proteins (nsp3, 4 and 12), spike proteins and nucleocapsid proteins following cold adaptation. Thus, the establishment of BP-caKII and the identified mutations in this study may provide insight into the genetic background of embryo and cold adaptations, and the attenuation of coronaviruses.

## 1. Introduction

Infectious bronchitis virus (IBV) is a pathogenic gamma coronavirus that causes respiratory symptoms, egg drops, nephritis and proventriculitis in domestic fowls [1]. Due to the persistent infection IBVs are difficult to eradicate and easy to mutate into new recombinant viruses in infected flocks [2,3]. The single-stranded, positive-sense RNA genome is approximately 27 kb and encodes RNA polymerase/transcriptase (1ab), spike (S), envelope (E), membrane (M), nonstructural (3a, 3b, 5a, 5b), and nucleocapsid (N) proteins [1]. Approximately 15 nonstructural proteins (nsp) are generated from the large 1a and 1ab proteins by viral proteases, and these nsp play roles in virus replication and pathogenicity [4,5]. The spike protein is a protective antigen that is essential for virus infection in host cells [1].

In Korea, several genotypes of IBV, including K-I, KM91-like, New cluster 1 (NC1), QX-like and recurrent QX-like, have been reported [6,7,8]. Among them, nephropathogenic KM91-like viruses were first reported in 1991 and became prevalent in the field. An inactivated oil emulsion vaccine and an embryo-adapted, attenuated live vaccine (K2 strain) have been used to prevent KM91-like virus infection, but occasionally, KM91-like viruses have been isolated in the field [8].

Cold adaptation has been used for the attenuation of various viruses, and a cold-adapted influenza A virus (IAV) has been successfully applied to the generation of attenuated human vaccine strains [9,10]. Cold-adapted IBV strains have also been established and characterized; however, the biological traits and genetic backgrounds of these cold-adapted IBVs have not been completely elucidated [9,10,11,12]. IBVs replicate firstly in the trachea then disseminate into internal organs. The temperature of the trachea is lower than internal organs, and it may be valuable to characterize the traits of cold-adapted IBVs. Furthermore, in comparison with respiratory and kidney pathogenicity tests, reproductive organ pathogenicity tests have not been easy to perform due to the long periods of observation necessary for the sexual maturation of hens [6]. Therefore, a time-saving and reproducible animal model to test reproductive organ pathogenicity is required [13].

In this study, we established a cold-adapted IBV (BP-caKII), and we characterized its pathogenicity in embryos and chickens, tissue tropism, and persistence of infection. We applied a premature reproductive tract pathogenicity model to the differentiation of the pathogenicity of IBVs and performed comparative genomics to shed light on the genetic background of the embryo adaptation of K2 and the cold adaptation of BP-caKII.

## 2. Materials and Methods

### 2.1. Virus, Egg and Chicken

SNU9106 was isolated from pooled tissues of the cecal tonsil and trachea of commercial layers (104-day-old) and was sent to the Laboratory of Avian Diseases of Seoul National University for diagnosis in 2009. A live attenuated commercial vaccine strain, K2 was previously established by 172 times passages of embryonated eggs [8]. A nephropathogenic field strain, KM91 that had been passaged 5 times through embryonated eggs was kindly provided by Avian Disease Division, animal and Plant Quarantine Agency in Korea. The viruses were propagated by inoculating into 10-day-old SPF embryonated eggs (Valo Biomedia, Adel, IA, USA) via the allantoic cavity route and incubated for 48 to 72 h, after which they were chilled at 4 °C overnight. The allantoic fluid was harvested, and the supernatant was stored at −70 °C after centrifugation at 3000 rpm (1915× *g*) for 10 min.

### 2.2. Primers, RNA Extraction, Real-Time and Conventional RT-PCR, and Genome Analysis

A primer set for real-time RT-PCR was designed based on the conserved region of the *nsp* 3 genes of 14 reference strains in the GenBank (Table 1). The reference strains used for the primer design are as follows: LX, Peafowl/GD/KQ6, KM91, QIA-03342, QIA-KR/D79/05, QIA-Q43, SNU-9106, SNU-10043, M41, Turkey coronavirus, SNU-8067, ITA/90254/2005, and ArkDPI11. Their accession numbers are available in Figure 3. Primers for genome amplification and sequencing were described previously and additional new primers were summarized in Table 1 [6].

Viral genomic RNA was extracted from 100 µL of infectious allantoic fluid using the Viral Gene-Spin Kit (iNtRON Biotechnology, Seongnam, Korea). SYBR real-time one-step RT-PCR (real-time RT-PCR) was performed using an Applied Biosystems StepOne real-time PCR machine and a one-step RT-PCR kit (KAPA-Biosystems, Boston, MA, USA) according to the manufacturer’s instructions. Total reaction volumes were adjusted to 10 µL and 1 µL of the extracted RNA used.

Conventional RT-PCR for genome amplification was performed using a one-step RT-PCR kit (Qiagen GmbH, Hilden, Germany) according to the manufacturer’s instructions. The RT-PCR conditions were as follows: cDNA synthesis at 50 °C for 30 min, inactivation of the reverse transcriptase at 95 °C for 15 min, 40 cycles of denaturation at 94 °C for 30 s, annealing at 50 °C for 30 s, and extension at 72 °C for 2 min, with a final extension at 72 °C for 5 min. The amplicons were purified using a PCR purification kit (MEGA-quick-spin Total Fragment DNA Purification Kit, iNtRON Biotechnology) and sequenced with PCR and sequencing primers using an ABI3711 automatic sequencer (Macrogen Co., Seoul, Korea). The overlapping gene fragments were assembled to obtain a single complete genome sequence using ChromasPro version 1.5 (Technelysium Pty Ltd., Brisbane, Australia). Nucleotide (nt) and amino acid (a.a) identity estimates and amino acid translations were obtained using BioEdit (ver. 5.0.9.1.). The full genome sequences of BP-caKII and K2 were submitted to the GenBank database. Phylogenetic analyses of the complete *S1* gene of the IBV strains were conducted using MEGA software (ver. 5.0.5, neighbor-joining method with Tamura-Nei distance and 1000 repeats of bootstrapping) [14].

### 2.3. Cold Adaptation of the Virus

SNU9106, propagated via three blind passages, was diluted by 10^−3^ and inoculated into 10-day-old SPF embryonated eggs (Valo Biomedia, Adel, IA, USA) via the allantoic cavity route; the embryonated eggs were incubated at 32 °C for 48 h, after which they were chilled at 4 °C overnight. The allantoic fluid was harvested, and the supernatant was stored at −70 °C after centrifugation at 3000 rpm (1915× *g*) for 10 min. The allantoic fluid harvested from the inoculated embryonated eggs was used for the real-time RT-PCR. The allantoic fluid with the lowest Ct value was inoculated into the embryonated eggs, and the same procedure was repeated 20 times to become BP-caKII.

### 2.4. Virus Titration, Embryo Pathogenicity and Embryo Mortality Rate by Virus

To test virus titer and embryo pathogenicity, BP-caKII was inoculated into 10-day-old SPF embryonated eggs (Valo Biomedia) via the allantoic cavity route and incubated at 37 °C for 48 h. The infected allantoic fluid was harvested after chilling at 4 °C overnight. Then, BP-caKII and K2 titers were determined by inoculating 10-fold serial dilutions (10^−1^–10^−7^) of the virus into five 10-day-old SPF embryonated eggs via the allantoic cavity. The inoculated embryonated eggs were observed for death and dwarfism for 5 days at 37 °C, and the 50% chicken embryo infectious dose (EID_50_/mL) was calculated using the Reed-Muench formula [15]. The mortality rate of BP-caKII and K2 was measured by mean death time (MDT), and absolute lethal dose (ALD). MDT was measured as previously reported [16,17,18]. Briefly, BP-caKII and K2 were diluted within the range of 10^2^ to 10^4^ EID_50_/100 µL and 100 µL was injected into 10-day-old SPF embryonated eggs via the allantoic cavity. Five embryonated eggs per dilution were inoculated at 8 a.m., and inoculation was repeated at 4 p.m. with the same dose. The candling of embryonated eggs for mean death time was performed at eight-hour intervals within one day. ALD was determined by the highest dilution at which all embryos died.

### 2.5. Growth Kinetics

BP-caKII and K2 diluted to 1 × 10^2^ EID_50_/100 µL were injected into 10-day-old SPF embryonated eggs (Valo BioMedia, Osterholz-Scharmbeck, Germany) via the allantoic cavity route and incubated at 32 °C and 37 °C for 60 h. After 0, 12, 18, 24, 30, 36, 48 and 60 h, three eggs inoculated with each virus were chilled at 4 °C overnight. The allantoic fluid of all the eggs was harvested and viral RNA was extracted for the real-time PCR. The Ct values of samples were measured, and the ΔCt was calculated by subtract Ct value of each sample from that of the 0 h sample. In addition, we performed regression analysis with Ct values of 2-fold diluted BP-caKII and K2 viruses and fitted linear equations of BP-caKII and K2. Theoretically 1 unit of ΔCt is the 2-fold difference of RNA copies, but we calculated the real fold difference of RNA of each virus by using the slope value of the linear equation (BP-caKII, *y* = 0.259*x* + 9.94, *R*^2^ = 0.997, 1.81-fold difference/Ct; K2, *y* = 0.267*x* + 11.1, *R*^2^ = 0.997, 1.84-fold difference/Ct). The independent experiment was repeated and each sample was tested in triplicate in an experiment.

### 2.6. Tissue Tropism and Persistence Test

To determine tissue tropism and the persistence of IBV strains, we performed experimental viral infection using 80 7-day-old SPF chicks that were assigned to 4 groups (*n* = 20 per group). All chicks were treated with PBS (negative control), BP-caKII, K2, or KM91 (1.5 × 10^5^ EID_50_/100 µL/chick) via ocular and intranasal routes, and observed for clinical signs and mortality for 4 weeks after the challenge. Each week, five chicks from each group were euthanized by cervical dislocation, and the pathological lesions of internal organs were observed throughout necropsy. The tracheal, cecal tonsil and kidney tissue samples were collected and examined using the real-time RT-PCR without virus titer calculation.

### 2.7. Generation of a Premature Reproductive Tract Pathogenicity Model Using DES (Diethylstilbestrol)

To investigate the influence of IBV on the premature reproductive organs of the chicks, we induced the sexual precocity by DES (Sigma) treatment [13]. A total of 48 1-day-old SPF female chicks (BioPOA Co., Yongin, Korea) were distributed into four different groups. Three groups were inoculated with BP-caKII, K2, and KM91 (10^6.5^ EID_50_/100 µL/chick) via ocular and intranasal routes, and a DES-control group was not inoculated with virus. A total of 12 birds were assigned to each group. After inoculation with each virus, the dead chicks were excluded when comparing reproductive tract lesions. At 5, 8 and 11 days after virus infection, all chicks were inoculated with DES intramuscularly (1 mg/100 µL) and were observed for clinical signs and mortality for 2 weeks after the challenge and were euthanized by cervical dislocation at 2 weeks post-infection. During the experiment, all chicks were reared in air-filtered isolators (THREESHINE, Daejeon, Korea) and feed and water were provided ad libitum. Gross lesions of the reproductive tract were examined based on cyst formation (four-grade-scoring: Negative, moderate, + (cyst covering less than 20% of oviduct), marked, ++ (covering 20% to 50%), and severe, +++ (covering more than 50% of oviduct)), and the presence or absence of caseous material in the oviduct.

### 2.8. Statistical Analysis

The significance of growth kinetics and embryo pathogenicity were evaluated by two way analysis-of-variance (ANOVA) and a Man-Whitney test, respectively. The difference in virus frequency in tissues and lesions in the premature reproductive tract pathogenicity model was assessed using chi-squared distribution (95% confidence intervals).

### 2.9. Ethical Statement

All animal experiments were performed at BioPOA Co. (Suwon, South Korea) following a protocol that adhered to the National Institutes of Health’s Public Health Service Policy on the Humane Care and Use of Laboratory Animals. The protocol was reviewed and approved by the Institutional Animal Care and Use Committee (IACUC) of BioPOA Co. (tissue tropism and persistence test, identification code: BP-2016-003-2; date of approval: 15 January 2016; premature reproductive tract pathogenicity test, identification code: BP-2016-001-2; date of approval: 15 January 2016).

## 3. Results

### 3.1. Establishment and Growth Kinetics of a Cold-Adapted IBV, BP-caKII

SNU9106 was passaged 20 times through three embryonated eggs by inoculating allantoic fluid with high virus titer, and BP-caKII was established. Out of the three allantoic fluids of each passage, the allantoic fluid with the highest virus titer after the real-time RT-PCR was selected for the next passage. The growth kinetics of BP-caKII in embryonated eggs at 32 °C and 37 °C were compared with those of K2 (Figure 1). According to our results, BP-caKII grew to higher titers at 37 °C than at 32 °C for the first 24 h, but then this result was reversed for the last 24 h. However, K2 showed significantly higher titers at 37 °C than at 32 °C and after 24 h. Therefore, BP-caKII showed a tendency to grow more efficiently at 32 °C than at 37 °C, and BP-caKII grew to a higher titer than K2 at 32 °C.

### 3.2. Embryonic Pathogenicity of BP-caKII

IBVs tend to acquire increased embryonic pathogenicity during embryonated eggs passages, and we compared the embryonic pathogenicity of BP-caKII with that of K2. BP-caKII and K2 showed 10^4.0^ ALD and 10^3.0^ ALD, respectively, and BP-caKII showed higher ALD than K2. In addition, the standard deviation (±SD) of MDT of BP-caKII was 106.4 ± 29.7 h, which is slightly longer than the 63.2 ± 15.3 h of K2 (*p* < 0.001). BP-caKII showed significant difference from K2 in MDT, but not in mortality (Table 2).

### 3.3. Tissue Tropism and Persistence of BP-caKII

The tissue tropism and persistence of BP-caKII have been compared with those of K2 and KM91 strains via the detection of viral RNA in the trachea, kidneys and cecal tonsils of infected and uninfected SPF chickens at 4 weeks of age. Except for the negative control group, the BP-caKII, K2- and KM91-inoculated groups showed different positive rates over the time period of observation (Table 3). BP-caKII, K2 and KM91 showed 1/5–3/5, 2/5–5/5 and 4/5–5/5 positive rates in chickens, respectively, during the first 3 weeks, which disappeared at 4 week-post-inoculation (WPI) in the trachea samples. In the kidneys, BP-caKII was detected in 3 out of 5 chickens (1–2 WPI) only at 3 WPI, but K2 and KM91 were detected in 2–3 chickens (2 WPI) at 2–3 WPI and 4–5 chickens at 1–3 WPI, respectively. In cecal tonsils, BP-caKII was detected in 3 out of 5 chickens only at 4 WPI, but K2 and KM91 were detected in 1–2 chickens at 2–4 WPI and 4–5 chickens at 1–4 WPI, respectively. Only KM91 showed significantly different positive rates from K2 and BP-caKII in the half of cases (*p* < 0.05).

### 3.4. Pathogenicity of BP-caKII in the Premature Reproductive Tract Pathogenicity Model

The premature reproductive tract pathogenicity model was used to compare the pathogenicity of BP-caKII, K2 and KM91, and all the chicks treated with DES showed a premature phenotype in the reproductive tract. (Table 4, Figure 2). Some chicks in all groups died during the experiment due to weakness or viral infection, and only the surviving chicks were included for lesion scoring. BP-caKII-infected chicks did not show any cyst or caseous material in the oviduct, but 11% (1/9) of K2-infected and 64% (5/8) of KM91-infected chicks did show the lesions. Only the frequency of lesions in the KM91 challenge group was significantly higher than that in other groups (*p* < 0.05).

### 3.5. Genome Sequence Analysis of BP-caKII and K2

To understand the genetic background of the cold adaptation and the decreased pathogenicity of BP-caKII, we determined the complete genomic sequence of BP-caKII (MF924724) and performed phylogenetic analysis with the *S1* gene. Unexpectedly, we found that BP-caKII was classified into the KII genotype (Figure 3), which was clearly different from the NC I genotype of the parent strain, SNU9106 [19]. Therefore, we determined the complete genome sequences of E5-, E10- and E15-passaged viruses and compared their amino acid sequences together with BP-caKII and SNU9106, using KM91 as a reference [20]. Out of the different amino acids in KM91, the amino acids that were different between the passaged viruses are summarized in Appendix A. Only one or two amino acids changed during E5–E20 passages, and most of the changes had already been acquired at the E5 passage.

For comparative genomics study, we determined the genomic sequence of K2 (MF924725). KM91 was used as a reference, and the length and GC content of the genomes, different nucleotide sequences in the 5′- and 3′-UTRs, and different amino acid sequences within each coding gene are summarized in Table 5 and Table 6. The length of the genomic sequence of BP-caKII was 3 and 6 nucleotides shorter than that of K2 and KM91 due to the deletions of 2 codons, F23del in S and I1839del in nsp 3. The GC content was very similar across viruses, ranging from 38.12% to 38.19%. The amino acids that differ between KM91 and both K2 and BP-caKII are summarized, and the different mutations between K2 and BP-caKII strains are represented in bold in Table 5 and Table 6. In comparison with KM91, K2 showed no amino acid changes in 3a, 3b, E, M, nsp 6, nsp 9 and nsp 10, but it did show multiple amino acid changes in other genes, especially *S*, *nsp13*, *nsp14* (98.4%), *nsp15* (97.3%) and *nsp16* (97.8%) (Table 6). In comparison with K2, BP-caKII shared a similar or single amino acid different sequence for nsp2, nsp7, nsp8, nsp13, nsp14, nsp15, nsp16, 4b, 4c, 5a, E, and M. Therefore, BP-caKII and K2 shared embryo-adaptation-related multiple amino acid changes in nsp13, 14, 15, 16, and S. However, BP-caKII also showed multiple amino acid changes in S, N, nsp3, nsp4, and nsp12 (Table 6). All of the mutations were unreported previously, and their effects on protein function were unknown. We summarize the location of each mutation in the functional domain, subunit, or specific region in Table 5 and Table 6.

## 4. Discussion

Passages of pathogenic IBVs through embryonated eggs resulted in virus attenuation and efficient virus replication, and all of the attenuated IBV vaccine strains, such as H120, K2 and YX10p90, have been established by embryonic passages [8,21,22]. Cold adaptation was another virus attenuation method, and a cold-adapted IAV vaccine strain has been used to generate vaccine strains against seasonal flu [9,10]. In a previous study, cold adaptation of IBV was reported, but the attenuation of pathogenicity was incomplete, and the genomic backgrounds of the attenuated strains were unavailable [11]. In this study, we established a cold-adapted, attenuated IBV strain, BP-caKII, and characterized its pathobiological and genomic traits.

Although we started embryo passages with a SNU9106 isolate that had been classified into the new cluster I genotype, the E5, E10, E15 and BP-caKII strains were classified into the KM91-like genotype. SNU9106 is a recombinant virus possessing the *1ab* gene of the KM91-like genotype and a recombinant *S1* gene containing a partial segment from the QX-like virus [20]. Therefore, BP-caKII was unlikely to have originated from SNU9106. Although the K2 strain has been used commercially since 2010, it might have been under clinical evaluation in the field for governmental approval for commercial use during this period, including 2009 [23]. The multiple shared mutations between K2 and BP-caKII, and the fact that both strains shared almost the same mutational patterns following cold-passaging (20 times), indicate that the K2 and BP-caKII strains may be closely related.

Although BP-caKII showed less pathogenicity in embryonated eggs than K2 they showed more delayed viral dissemination to the kidneys and cecal tonsils than did KM91. KM91 was detected and persisted in the trachea and kidneys for the first 3 weeks and in the cecal tonsils throughout the whole period of observation (4 weeks). Pathogenic field isolates of QX-like and variant IBVs also persisted for a long time in the trachea, proventriculus, kidneys and cecal tonsils of infected chickens [24]. Therefore, embryo and cold adaptation apparently decreased pathogenicity and changed the tissue tropism of K2 and BP-caKII in embryonated eggs and chickens. However, in certain conditions, even vaccine strains may persist for a long time before being excreted into the trachea and cloaca [3]. Thus, vaccine strains cleared within a short period of time, but with enough time to induce protective immunity, may be preferable to minimize the conversion of pathogenicity and the occurrence of recombinant viruses.

To date, various pathogenicity models of IBVs have been developed, but testing pathogenicity on reproductive organs requires a long period before the sexual maturation of hens occurs [6]. DES has been known to induce abnormalities in female reproductive tracts and to induce a premature phenotype in the reproductive tracts of female chicks [13,25]. Previously, the premature reproductive tract pathogenicity model was established and used to evaluate IBV pathogenicity [13]. In this study, we verified the pathogenicity of BP-caKII, K2 and KM91 by using this model. KM91 showed a significantly higher pathogenicity than BP-caKII and K2, but the pathogenicity of BP-caKII was insignificantly different from K2. Considering the differences in pathogenicity in embryos and premature reproductive tracts, as well as the differences in tissue tropism, BP-caKII may be less pathogenic than K2.

The comparative genomics study of BP-caKII and K2 revealed mutations acquired during embryo and cold adaptation. K2 acquired relatively more missense mutations in nsp13, nsp14, nsp15 and *nsp16* genes than other genes during embryo adaptation. Nsp13 is multifunctional, and its RNA 5’ triphosphatase and helicase domains are located on its N- and C-terminals, respectively [26,27]. The helicase domain of nsp13 is subdivided into the RecA1 and RecA2 domains, and four out of the five mutations identified were located in RecA2 [26]. Nsp14 is composed of the N-terminal exonuclease (ExoN) and C-terminal AdoMet-dependent guanosine N7-methyltransferase (N7-MTase) domains, and all the mutations were located in both domains [28,29,30]. ExoN and N7-MTase play roles in the correction of replication errors and in mRNA capping, respectively [28,29]. Nsp16 is an AdoMet-dependent 2’-O-methyltransferase (2’-O-MTase) that plays a role in the capping of viral RNAs, which helps in the evasion of innate immunity [31]. Nsp15 is an endoribonuclease that plays an important role in the evasion of dsRNA sensors, such as RIG1 and MDA5, in infected cells [32]. The mutations shared by K2 and BP-caKII in the *nsp14, 15*, and *16* genes may facilitate or impair innate immunity evasion in embryos and chickens, respectively. However, the exact innate immunity differences between embryos and chickens have never been elucidated. To date, embryo-adapted strains of IBVs have shown high mutation rates in the *nsp3* genes, including a large nucleotide deletion in the 3’ end of the *N* gene and partial coding region of gene 6 [22,33]. Therefore, mutations for embryo adaptation may be strain-specific.

BP-caKII acquired multiple additional missense mutations in the *nsp 3*, *4*, *12*, *S* and *N* genes during cold adaptation, but the exact functions of these mutations were unknown. Enzyme activity may be affected at different temperatures. *Nsp3* has multiple domains, and three out of the four mutations were located in the PL (papain-like protease, L978F) and 3Ecto (NSP3-ectodomain, I1167del and V1168I) domains [34]. Nsp12 is an RNA-dependent RNA polymerase (RdRp), and the P214L mutation and the R626C and P812I mutations are located in the Nidovirus RdRp-associated nucleotidyltransferase (NiRAN) and the RdRp domains, respectively. *Nsp4* is involved in the formation of the replication and transcription complex (RTC) with nsp3, and nsp12 is an essential component of the RTC. Therefore, further study on the roles of the multiple mutations of nsp3, 4 and 12 seen during viral replication at the lower temperature may be interesting. The R118M mutation in S1 NTD of K2 and BP-caKII is located in the partial ceiling region of the receptor binding site, and V66A of BP-caKII is located in the vicinity of the R118M mutation site [35]. Evolutionarily, coronaviruses developed a ceiling to protect the receptor binding site in S1 NTD from host immune surveillance, and the R118M and V66A mutations may be related to the function of the partial ceiling [35]. The accumulation of multiple mutations in the *N* gene was interesting, but their functions in cold adaptation are unclear. The cold-adapted IAV showed mutations in its nucleoprotein, as well as the polymerase genes [36]. The RNA binding domain and SR (serine and arginine-rich) region of N interact with the ubiquitin-like domain of nsp3, and K136T and I204V mutations are located in the NTD (N-terminal domain-a.a 19–162) RNA binding site (136–190) and LKR (Linker region) (a.a 163–218), respectively [37,38]. The V317L and T343S mutations are in the putative dimerization core (a.a 250–349) of CTD (C-terminal domain-a.a 219–349) that binds single-stranded RNA [39]. Therefore, multigenic mutations may be acquired during cold adaptation of coronavirus.

## 5. Conclusions

In conclusion cold-adaptation of an IBV strain increased growth efficiency at 32 °C decreased pathogenicity in embryos further, and different multiple genes may be involved in embryo- and cold-adaptation of IBV.

## Figures and Tables

**Figure 1 viruses-10-00652-f001:**
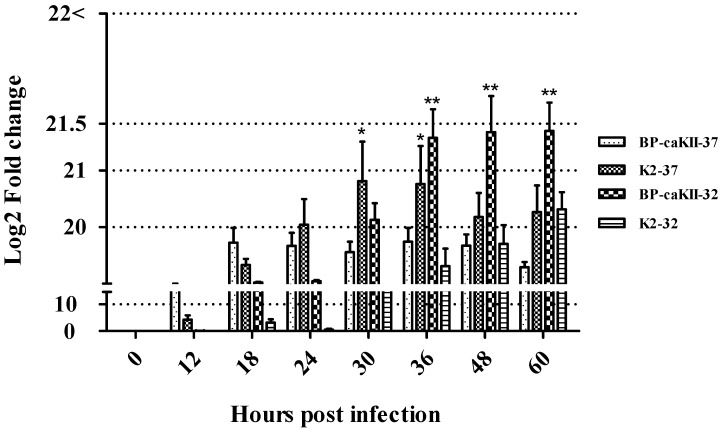
The growth increment of BP-caKII and K2 in embryonated eggs at 32 °C and 37 °C, over time (hour, h). The time point and the mean fold change of RNA copy (log2) were represented on the *x*- and the *y*-axis, respectively. Fold changes were calculated by multiplying the ddCt value and real fold-change of each virus calculated from regression analysis. * Significant difference of K2-37 °C from BP-caKII-37 °C and K2-32 °C; ** significant difference of BP-caKII-32 °C from others (*p* < 0.05).

**Figure 2 viruses-10-00652-f002:**
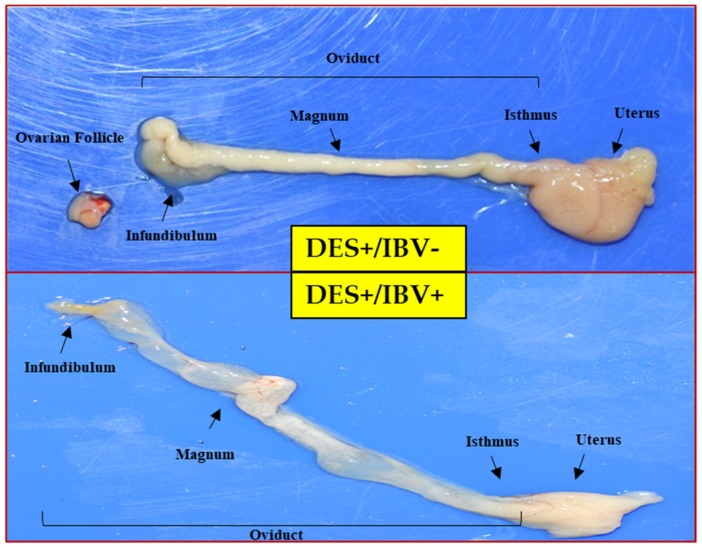
Differences in pathological changes between IBV strains according to sexual precocity after IBV infection. The severe aplasia of the ovarian follicles and atrophy of the oviduct was compared across treated (+) and untreated (−) chicks. The yellow arrow indicates a lesion, such as a cyst, in reproductive organs.

**Figure 3 viruses-10-00652-f003:**
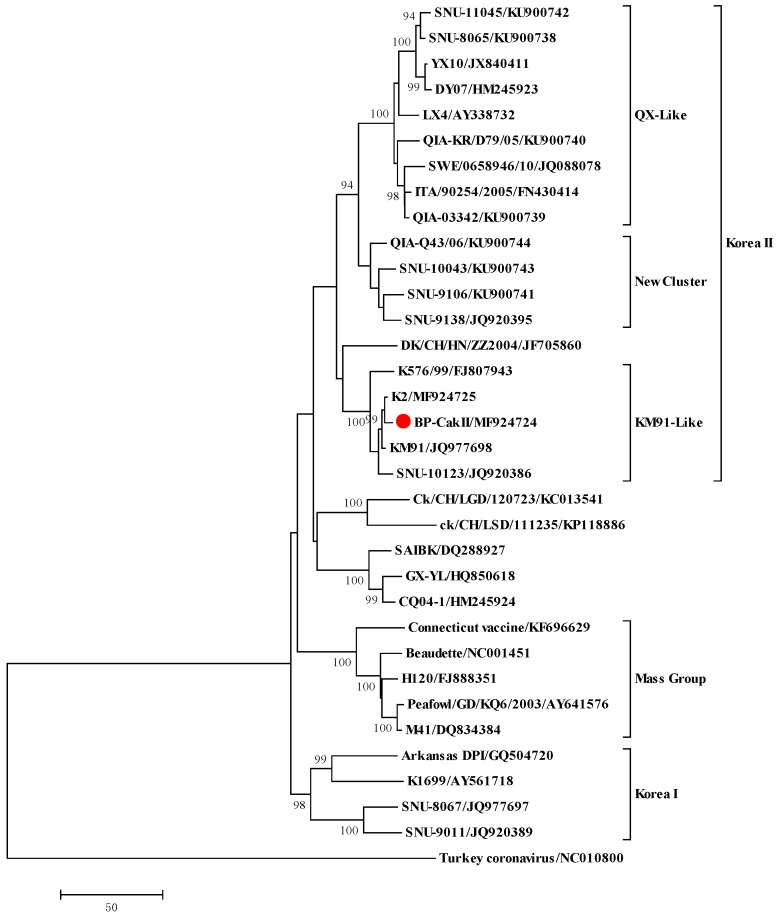
Phylogenetic analysis of the *S1* genes of infectious bronchitis viruses. Phylogenetic trees are based on the S1 amino acid sequence of IBVs, where the BP-caKII strain is marked with a red filled circle. Phylogenetic trees were constructed with the neighbor-joining method using MEGA 6.0. The bootstrap values were determined from 500 replicates of the original data. The branch number represents the percentage of times that the branch appeared in the tree. Bootstrap values greater than 50% are shown. The p-distance is indicated by the bar at the bottom of the figure.

**Table 1 viruses-10-00652-t001:** Primer sets used for real-time RT-PCR and full genome sequencing.

Primer	Amplicon (bp)	Forward (5′ to 3′)	Reverse (5′ to 3′)	Usage
*Nsp* 3	158	CACCTACAAGTTAACACCTG	GTAGGAATGCGAAGAGACTTAC	Real-Time RT-PCR
GS1	1487	ACTTAAGTGTGATATAAATATATATC	GTTTGGTCAAGCAGTGTTAGG	Sequencing
GS2	1580	GTCAATTGTTGTCCTAGCAGC	CACATGTAGCTGGTTTAAC	
GS3	755	GAATTCATGGAGACTTGCTCTTC	GTCTTGTATAAGAGCCAACAC	
GS4	471	TTCTGATGTTCCTAGAGAAG	CGCCATCTACAAGAACATTC	
GS5	598	CCGCTCTGTTGTTGTAAAAC	GGGCAATTTGAATATTGCGTC	
GS6	1629	CTGGACTGGTTTGTTCAAAC	AAGACAATGGTCGCATAAGC	
GS7	1059	GGAAGCATTGAAATGTGAAC	GGAATGTACCAAGGTTTTCGC	
GS8	760	TCTCACTGCCTAAGTGGTTG	CCTTCTGTATATGCAGTAAG	
GS9	551	CACATACCATCTTATGCTG	CATGCTACATTATCACCAC	
GS10	543	CTTACAGTCTAAAGGGCATG	ATCAGGATCACATCCACTAGC	
GS11	501	AGCGAGCCTTTGATGTATG	GGTTTCCGAACTCAATAGC	
GS12	1713	CACTGCATGTTCTCATGCAGC	ACAACGCGTCATTATAGCATC	
GS13	2159	CAATCCGGAATTGGAACAG	CTTAGCCTTAGTAATGCGAG	
GS14	1441	GCTACCTAACACACTAAACAC	GCACTTTGGTAGTAGTACAC	
GS15	616	TGGTCCTGTCTGTGATAAC	CGTAAGAATAGCACTCTGC	
GS16	363	GGTAGTGGAAGACATGTTC	CCACCATTTTGACAACTCGTC	

**Table 2 viruses-10-00652-t002:** Comparison of embryonic pathogenicity of BP-caKII with K2.

Virus	Titer (EID_50_/0.1 mL)	Hours Post-Infection														
		40	48	64	72	80	88	96	104	112	120	136	144	Mortality	ALD ^a^	MDT ^b^
BP-caKII	1 × 10^4.0^	-	1	-	-	1	1	1	1	1	1	2	1	100%	1 × 10^4.0^	106.4 ±29.7 * (hours)
	1 × 10^3.0^	1	-	-	1	1	1	1	2	-	1	-	-	80%		
	1 × 10^2.0^	-	-	-	1	-	-	1	-	-	-	-	-	20%		
K2	1 × 10^4.0^	2	1	2	3	2	-	-	-	-	-	-	-	100%	1 × 10^3.0^	63.2 ±15.3 (hours)
	1 × 10^3.0^	1	-	-	1		2	2	2	-	-	2	-	100%		
	1 × 10^2.0^	-	-	-	2			1	-	-	-	-	-	30%		

^a^ ALD: Absolute lethal dose (the lowest dose causing 100% mortality). ^b^ MDT: Mean death time. * Significant difference of standard deviation (±SD) from K2 (*p* < 0.05).

**Table 3 viruses-10-00652-t003:** Comparison of tissue tropism and persistency of infectious bronchitis viruses (IBVs).

	Positive Rate of Virus											
	Trachea				Kidney				Cecal Tonsils			
	1 ^a^	2	3	4	1	2	3	4	1	2	3	4
Cont	0/5	0/5	0/5	0/5	0/5	0/5	0/5	0/5	0/5	0/5	0/5	0/5
BP-caKII	3/5	4/5	1/5	0/5	0/5	0/5	3/5	0/5	0/5	0/5	0/5	3/5
K2	5/5	3/5	2/5	0/5	0/5	2/5	3/5	0/5	0/5	1/5	2/5	1/5
KM91	4/5	5/5	4/5	0/5	5/5 *	5/5 *	4/5	0/5	4/5 *	4/5 **	5/5 **	5/5 ***

^a^ Weeks-post-inoculation. * Significant difference from K2 and BP-caKII (*p* < 0.05). ** Significant difference from BP-caKII (*p* < 0.05). *** Significant difference from K2 (*p* < 0.05).

**Table 4 viruses-10-00652-t004:** Gross lesion scoring of reproductive organs of IBV by the diethylstilbestrol (DES)-treated oviduct model.

Group	Frequency of Pathological Lesions				
	Cyst Score ^a^			Caseous Material in the Oviduct	Total Lesion Frequency (%)
	+	++	+++		
DES.C	0/7	0/7	0/7	0/7	0/7 (0)
BP-caKII	0/11	0/11	0/11	0/11	0/11 (0)
K2	0/9	1/9	0/9	0/9	1/9 (11)
Km91	0/8	1/8	1/8	3/8	5/8 (64) ^b^

^a^ +, Moderate; ++, Marked; +++, Severe. ^b^ Significantly different from others (*p* < 0.05).

**Table 5 viruses-10-00652-t005:** Comparison of nucleotide and amino acid sequences of KM91 with K2 and BP-caKII.

Gene	Length of KM91/K2/BP-caKII	K2		BP-caKII	
		Identity (%, nt/aa)	Missense Mutations	Identity (%, nt/aa)	Missense Mutations
Genome	27,629 (38.12)/27,626 (38.19)/27,623 (38.13)	99.2%		99.1%	
***5′-UTR***		99.6%	G270A (nt)	99.4%	**T4G**, **G7A**, G270A
***1a***	3952	99.8%/99.7%	Table 6	99.8%/99.6%	Table 6
***1ab***	6630/6329	99.0%/99.0%	Table 6	98.9%/99.2%	Table 6
***S***	1163/1162/1162	99.6%/99.5%	**S1**:L2S, V21del, F23N, H24N, R118M, D406G, K488E **S2**: F697L, S883F	99.4%/98.6%	**S1**: L2S, V21del, F23N, H24N, **V66A**, R118M, **F273L**, **S365I**, K488E, **V500L**, **S2**: F697L, **I847F**, **S879T**, S883F, **S1006F**
***3a***	48	100%/100%		100%/100%	
***3b***	62	100%/100%		100%/100%	
***E***	109	99.6%/100%		99.6%/99%	**L27F**
***M***	226	100%/100%		99.7%/99.5%	**L80F**
***4b(ORF X)***	94	99.6%/98.9%	H21R	99.6%/98.9%	H21R
***4c***	56	100%/100%		99.4%/98.2%	**I36V**
***5a***	65	99.4%/98.4%	F7C	99.4%/98.4%	F7C
***5b***	82	99.5%/98.7%	Q66R	99.5%/98.7%	Q66R
***N***	409	99.5%/99.7%	S109R	99%/98.5%	**NTD**: S109R, **K136T LKR**: **I204V CTD**: **V317L**, **T343S IDR: N407Y**
***6b***	73	98.6%/97.2%	**Y29H**	99%/98.6%	V500G
***3′-UTR***	27,324/27,327	98.5%	**A89T**, A189G, **T206C**, **G275A**	98.5%	A189G, **G193A**, **A246G**, **G270A**

Different sequences between K2 and BP-caKII are represented in bold.

**Table 6 viruses-10-00652-t006:** Comparison of nonstructural proteins of KM91 with K2 and BP-caKII.

*NSP* (Size)	Putative Function ^b^	Protease ^a^	K2		BP-caKII	
			Identity	Missense Mutation	Identity	Missense Mutation
2 (673)	Unknown	PLP	99.5%	A30V, K320T, H478Y	99.5%	A30V, K320T, H478Y
3 (1593/1592)	Papain-like viral protease	PLP	99.8%	T189I(PL1), D1510A(YD)	99.6%	T189I (PL1), **P321L (PL1)**, **L978F (BSM)**, **V1167del (3Ecto) V1168I (3Ecto)**, D1510A(YD)
4 (514)	Unknown	PLP/3CLpro	99.6%	**W16R**, T50A	99.4%	T50A, **R417C**, **Q514H**
5 (307)	Coronavirus endopeptidase C30	3CLpro	99.6%	**T171P**	100%	
6 (293)	Hydrophobic domain	3CLpro	100%		100%	
7 (83)	nsp7 superfamily	3CLpro	98.7%	D43Y	97.5%	**S4I**, D43Y
8 (210)	nsp8 superfamily	3CLpro	99.5%	T155I	99.5%	T155I
9 (111)	nsp9 superfamily	3CLpro	100%		100%	
10 (145)	nsp10 superfamily, RNA synthesis	3CLpro	100%		100%	
11 (21)	RNA-dependent RNA polymerase	3CLpro	100%		100%	
12 (918)	RNA-dependent RNA polymerase	3CLpro	99.7%	S567N, E832D	99.4%	**P214L**, S567N, **R626C**, **P812L**, E832D
13 (600)	Viral RNA helicase	3CLpro	99.1%	S51L, **RecA2 domain**: I474V, P555Q, S559N, G594D	99.0%	S51L, **RecA2 domain**: I474V, **T512I**, P555Q, S559N, G594D
14 (521)	nsp11 superfamily; exoribonucelase	3CLpro	98.4%	**T207S**, L252Q, N256D, D287G, V455I, S505T, Q507N, N516S	98.6%	L252Q, N256D, D287G, V455I, S505T, Q507N, N516S
15 (338)	Nidoviral uridylate-specific endoribonucelase	3CLpro	97.3%	I22M, I38V, V133I, N173S, D202E, V249A, I265M, I297S, K308R	97.0%	I22M, I38V, V133I, N173S, D202E, V249A, I265M, I297S, K308R, **S313L**
16 (275)	23S rRNA methylase	3CLpro	97.8%	F110L, V143I, K160R, V173L, L209I, K270Q	97.8%	F110L, V143I, K160R, V173L, L209I, K270Q

^a^ PLP, papain-like protease; 3CLpro, 3C-like protease; Different sequences between K2 and BP-caKII are represented in bold.

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
