# Peer review of "Pathobiological and Genomic Characterization of a Cold-Adapted Infectious Bronchitis Virus (BP-caKII)"

_viruses, 2018, doi:10.3390/v10110652_

Round 1

Reviewer 1 Report

The authors have established a cold-adapted IBV (BP-caKII) by passaging a field virus through SPF eggs 20 times at 32°C. Then, they have characterized its growth kinetics compared to the parent virus K2, pathogenicity in eggs, it’s tropism and pathogenicity to reproductive tract using a newly established premature reproductive tract pathogenicity model. They are also providing the genetic basis for this cold adaptation. The work is novel and much required to the field. Some of the conclusions are not based on the findings and not based on valid data analysis. The limitations are

1)     Growth kinetics data (Fig 1): you calculated EID50 based on qPCR results and is it acceptable? qPCR detects both dead and live virus and I do not think you are able to talk about infectious dose here

2)     Fig: if you have done in triplicate please add error bars to the graph, then do analyze the data with proper stat analysis. Without stat analysis you cannot make statements such as written at line 177.

3)     Line 140: in fact you used 80 animals and divide into 4 groups (n=20 per group)

4)     Lines 147-160: you would have included another group without DES to rule whether the lesions in fact caused by the virus rather than DES action on immature reproductive tract. Did you isolate the virus from the reproductive tract? Did you look at histological changes? Did you do post mortem analysis of  the animals found dead to see whether they got reproductive lesions?

5)     Line 158-160: Describe the lesions scoring system properly. Did you score each segment? Did you score ovaries? Were there any other lesions in any other organ?

6)     Table 2: On what basis you say that BP-caKII showed less pathogenicity (Lines 190-191). The numbers for cold adapted and control virus are not very apart. You need to analyze this data using a stat method in order make statement.

7)     Table 3: Again the numbers for cold adapted and control virus are not very apart. You need to analyze this data using a stat method in order make statements. Also provide IBV genome data as well in the table.

8)     Table 4: Again the numbers for cold adapted and control virus are not very apart. You need to analyze this data using a stat method in order make statements.

9)     Fig 2: label different parts of the reproductive tract. Include ovaries as well.

10)   Line 85: Fig 2 is not showing such information

11)   Line 75: Country?

12)   Lines 345-347: Revise the conclusion following proper data analysis using a statistical method.

13)   Ref 13: Is it a thesis?

14)   ECFs: is not a good abbreviation. Say “embryonated eggs”

15)   Line 157: This company is making chicken specific air-filtered isolators?

16)   Animal use protocol all animals to die? Or you euthanized after determining the humane end point?

Author Response

Open Review 1

Comments and Suggestions for Authors

The authors have established a cold-adapted IBV (BP-caKII) by passaging a field virus through SPF eggs 20 times at 32°C. Then, they have characterized its growth kinetics compared to the parent virus K2, pathogenicity in eggs, it’s tropism and pathogenicity to reproductive tract using a newly established premature reproductive tract pathogenicity model. They are also providing the genetic basis for this cold adaptation. The work is novel and much required to the field. Some of the conclusions are not based on the findings and not based on valid data analysis. The limitations are

1)     Growth kinetics data (Fig 1): you calculated EID50 based on qPCR results and is it acceptable? qPCR detects both dead and live virus and I do not think you are able to talk about infectious dose here

Response 1: We followed the reviewer’s opinion and we compared fold difference of RNA copies of samples instead of the converted EID50 (lines 136-140).

2)     Fig: if you have done in triplicate please add error bars to the graph, then do analyze the data with proper stat analysis. Without stat analysis you cannot make statements such as written at line 177.

Response 2: We changed the Fig. 1 with bar chart for better explanation of the result and added the results of stat analyses as recommended.

3)     Line 140: in fact you used 80 animals and divide into 4 groups (n=20 per group)

Response 3:We changed ‘twenty’ into ‘eighty’ (line 143) and added ‘(n=20 per group)’ you suggested.

4)     Lines 147-160: you would have included another group without DES to rule whether the lesions in fact caused by the virus rather than DES action on immature reproductive tract.

Response 4: Previously Prof. Jae-Hong Kim confirmed that only virus, not DES, caused lesions (Kim, 1995). However, we included the DES negative group but during the experiment 7 chicks were dead due to weakness. Usually we select healthy chicks by choosing average-weight or over-average-weight chicks for experiment. But we used all SPF female chicks (including under-average-weight chicks) after chick sexing we have no room to select healthy chicks due to reduction of experimental animals (3R principle). Only 3 chicks were alive but all the 3 chicks showed no lesions and we did not include the result in the manuscript.

Did you isolate the virus from the reproductive tract? Did you look at histological changes? Did you do post mortem analysis of the animals found dead to see whether they got reproductive lesions?

Response 4-1: No, we didn’t. We only focused on lesions of prematured reproductive organs of live chicks.

5)     Line 158-160: Describe the lesions scoring system properly.

Response 5: We described the lesion scoring system in detail in lines 162-164.

Did you score each segment?

Response 5-1: No, we scored whole oviduct without considering segmental location (there is no selective occurrence of cyst in specific segments) (lines 162-164).

Did you score ovaries?

Response 5-2:No, because we couldn’t find maturation of follicles and gross lesions in ovaries.

Were there any other lesions in any other organ?

Response 5-3:We focused only on reproductive tract.

6)     Table 2: On what basis you say that BP-caKII showed less pathogenicity (Lines 190-191). The numbers for cold adapted and control virus are not very apart. You need to analyze this data using a stat method in order make statement.

Response 6:We performed statistical analysis and found that the MDT of BP-caKII was significantly different from K2. We revised line 194-195 and Table 2.

 7)     Table 3: Again the numbers for cold adapted and control virus are not very apart. You need to analyze this data using a stat method in order make statements. Also provide IBV genome data as well in the table.

Response : We performed statistical analysis and found no significant difference between K2 and BP-caKII. But KM91 showed significant difference from K2 and BP-caKII (lines 216-217).

8)     Table 4: Again the numbers for cold adapted and control virus are not very apart. You need to analyze this data using a stat method in order make statements.

Response 8:We performed statistical analysis and found that only significantly higher lesion frequency of KM91 than those of K2 and BP-caKII (lines 230-231).

9)     Fig 2: label different parts of the reproductive tract. Include ovaries as well.

Response 9: We labeled each parts of the reproductive organs.

10)   Line 85: Fig 2 is not showing such information

 Response 10: We revised fig 2 into fig3 (line 85). The GeneBank accession number can be found in the phylogenetic tree of Fig 3.  

11)   Line 75: Country?

Response 11: The animal and plant quarantine agency is in Korea. We added ‘Korea’, to line 75.

12)   Lines 345-347: Revise the conclusion following proper data analysis using a statistical method.

Response 12: We revised conclusion according to the results (lines 364-366). We also revised lines 302-305 according to the results.

13)   Ref 13: Is it a thesis?

Response 13: Ref 13 is a doctoral dissertation by Professor Jae-hong Kim, co-author of this study.

14)   ECFs: is not a good abbreviation. Say “embryonated eggs”

Response 14:We revised every ‘ECEs’ into ‘embryonated eggs’ in this paper

15)   Line 157: This company is making chicken specific air-filtered isolators?

Response 15: Yes, this company is an isolator maker (http://www.threeshine.com/).

16)   Animal use protocol all animals to die? Or you euthanized after determining the humane end point?

Response 16:We euthanized after determining the humane end point.

Reviewer 2 Report

Infectious bronchitis virus (IBV) is one of the major threats to poultry worldwide. Many modified live or inactivated vaccines are being used. However, like in the case of other coronaviruses (CoVs) recombination events led to the emergence of novel IBV strains whose virulence is unpredictable. This fact makes also difficult the eradication of IBV using current vaccines. In this manuscript, cold-adaptation and serial passaging in embryonated chicken eggs (ECEs) was used to generate an adapted IBV strain. Characterization of virus growth, pathogenesis and determination of viral sequence is included in the presented work. Nevertheless, some controls are missed and several aspects should be considered to strength the scientific significance of the manuscript.

Specific comments:

1. The main concern is that it could not be established whether the changes in pathogenesis, sequence, etc were derived for ECEs adaptation or cold-adaptation. In fact, the data on BP-caKII being more similar to K2 virus than to SNU9106 seems to point that ECEs adaptation was predominant over cold-adaptation.

To clarify this issue, ECEs (but not cold) adaptation of parental SNU9106 is an essential control missed in all experiments. To allow undoubtful conclusions, the experiments and comparisons should be performed using parental (virulent) SNU9106 virus, ECEs-adapted SNU9106 virus, and BP-caKII virus.

2. The rationale for performing cold adaptation instead of “traditional” virus passage in ECEs is not explained. Which are the advantages and disadvantages of cold adaptation compared with “regular” passaging? Why is cold adaptation selected in this case? Is wild-type IBV cold-sensitive?

In line 57 it is indicated that cold-sensitive IBV strains have been previously established and characterized. How did BP-caKII compare with those strains? Where is the novelty in the approach described in this manuscript?

3. Fig. 1. Is BP-caKII virus really cold-adapted? It is surprising that the virus grows better at 37 ºC than at 32 ºC. Control ECEs-adapted (37 ºC) SNU9106 virus is missing in this figure to convince that BP-caKII is cold-adapted.

4. There are two highly confusing sentences in the manuscript that should be clarified:

4.1. Line 187: “IBV tend to acquire increased embryonic pathogenicity during ECEs passages” Why is then ECEs passage used to obtain attenuated viruses?

4.2.  Line 278-279: “BP-caKII was unlikely to have originated from SNU9106” How do this fit with SNU9106 being used as parental strain to start virus passaging?

5. Development of a novel reproductive pathogenicity model is an important aspect of the work included in the manuscript. To validate the model, it would be helpful to compare the results obtained with this model and with a classical pathogenicity model, such as one analyzing respiratory symptoms.

Minor comments:

1. Page 2, line 84. Reference to Fig. 2 should not be there. In addition, GeneBank accession number mentioned should be included in a separate table or figure.

2. Line 226. “S1 gene” is not correct and should be replaced by “S gene sequence comprising S1 domain”

3. Reference 12 should include “Adv. Exp. Med. Biol. 494:557-62”

Author Response

Comments and Suggestions for Authors

Infectious bronchitis virus (IBV) is one of the major threats to poultry worldwide. Many modified live or inactivated vaccines are being used. However, like in the case of other coronaviruses (CoVs) recombination events led to the emergence of novel IBV strains whose virulence is unpredictable. This fact makes also difficult the eradication of IBV using current vaccines. In this manuscript, cold-adaptation and serial passaging in embryonated chicken eggs (ECEs) was used to generate an adapted IBV strain. Characterization of virus growth, pathogenesis and determination of viral sequence is included in the presented work. Nevertheless, some controls are missed and several aspects should be considered to strength the scientific significance of the manuscript.

Specific comments:

1. The main concern is that it could not be established whether the changes in pathogenesis, sequence, etc were derived for ECEs adaptation or cold-adaptation. In fact, the data on BP-caKII being more similar to K2 virus than to SNU9106 seems to point that ECEs adaptation was predominant over cold-adaptation.

To clarify this issue, ECEs (but not cold) adaptation of parental SNU9106 is an essential control missed in all experiments. To allow undoubtful conclusions, the experiments and comparisons should be performed using parental (virulent) SNU9106 virus, ECEs-adapted SNU9106 virus, and BP-caKII virus.

Response 1: We could not adapt SNU9106 virus to embryonated eggs by cold adaptation and the origin of BP-caKII may be K2 vaccine strain-related, embryo-adapted virus that had co-existed with SNU9106 in the sample (we explained possible origin of BP-caKII in the manuscript). The reason why we directly passaged already-embryonated egg-adapted K2 strain was to distinguish mutations acquired during cold-adaptation. By doing it we verified similar mutations were acquired by K2 as BP-caKII acquired. Thus, additional mutations acquired by BP-caKII may be associated with cold adaptation.

2. The rationale for performing cold adaptation instead of “traditional” virus passage in ECEs is not explained. Which are the advantages and disadvantages of cold adaptation compared with “regular” passaging? Why is cold adaptation selected in this case? Is wild-type IBV cold-sensitive?

Response 2: We added brief explanation on the reason why we select cold-adaptation in lines 59-61. Cold-adaptation of embryonated egg-adapted IBV may decrease embryo pathogenicity and internal organ tropism, but now we are neutral on whether cold-adaptation is good for vaccine establishment or not.

In line 57 it is indicated that cold-sensitive IBV strains have been previously established and characterized. How did BP-caKII compare with those strains? Where is the novelty in the approach described in this manuscript?

Response 2-1: We can’t directly compare with the previous cold adapted strain. However, BP-caKII is less pathogenic and more efficiently growing than K2 strain in terms of MDT and growth at 32°C, respectively. The previous work did not unravel the cold adaptation-related genetic changes but we did.

3. Fig. 1. Is BP-caKII virus really cold-adapted? It is surprising that the virus grows better at 37 ºC than at 32 ºC. Control ECEs-adapted (37 ºC) SNU9106 virus is missing in this figure to convince that BP-caKII is cold-adapted.

 Response 3: The previous Fig. 1 was not clear to explain our result, and we changed Fig. 1 according to the reviewer 1’s recommendation. BP-caKII grows significantly better at 32ºC than at 37 ºC, and more importantly BP-caKII grows significantly better than K2 at 32ºC. K2 grows significantly better than BP-caKII at 37ºC at 30hpi. Thus, BP-caKII may be cold-adapted.

4. There are two highly confusing sentences in the manuscript that should be clarified:

4.1. Line 187: “IBV tend to acquire increased embryonic pathogenicity during ECEs passages” Why is then ECEs passage used to obtain attenuated viruses?

Response 4.1: We also agree to reviewer’s opinion and that is our question. Field strain of pathogenic IBV showed no mortality or embryonic lesions (dwarfism) but after several passages virus titer increases and causes dwarfism (curled embryo). In case of commercial vaccine strains, K2 and H120 (passaged 190 and 160? times through embryonated eggs) showed increased pathogenicity in embryos but less pathogenicity in chicks. We hope to answer the question in the future and our mutation data may be useful for the work.

4.2.  Line 278-279: “BP-caKII was unlikely to have originated from SNU9106” How do this fit with SNU9106 being used as parental strain to start virus passaging?

Response 4-2:SNU9106 might have been major population but it could not have adapted to cold embryos.

5. Development of a novel reproductive pathogenicity model is an important aspect of the work included in the manuscript. To validate the model, it would be helpful to compare the results obtained with this model and with a classical pathogenicity model, such as one analyzing respiratory symptoms.

Response 5:BP-caKII was cold-adapted virus and we intended to compare its trait at higher temperature. So, we selected model (kidney tropism) reflecting rather higher temperature condition of kidney than lower temperature condition of trachea.

Minor comments:

 1. Page 2, line 84. Reference to Fig. 2 should not be there. In addition, GeneBank accession number mentioned should be included in a separate table or figure.

Response 1:We revised fig 2 into fig3 (line 85). If possible, we hope not to add an additional Table for accession no. 

2. Line 226. “S1 gene” is not correct and should be replaced by “S gene sequence comprising S1 domain”

Response 2:We revised it as recommended.

3. Reference 12 should include “Adv. Exp. Med. Biol. 494:557-62”

Response 3:We added it as recommended. 

Round 2

Reviewer 1 Report

The authors have addressed the comments satisfactorily. But I am seeing some new issues that are minor.

1.     All the tables: Column heading should be bold and better to separate the column heading with horizontal lines

2.     Table 1: Do you need a last column, “usage” since you indicated the use in your table title?

3.     Table 3: Center the titles over the appropriate columns (Trachea, Kidney,  Cecal tonsils)

4.     Fig 1: X axis title indicates “hours” and, after each number it is not necessary to specify “h”

5.     Fig 1 Y axis: what do you mean by “Mean fold change of RNA copy/log2”?

6.     Fig 1: You do not need a chart title since you have a figure legend with a title

7.     Fig 2: Label  all the components of the reproductive tract in both IBV negative and IBV positive tracts. Consistently label the panels (IBV infected and uninfected), either on the top or bottom.

Author Response

The authors have addressed the comments satisfactorily. But I am seeing some new issues that are minor.

1.     All the tables: Column heading should be bold and better to separate the column heading with horizontal lines

Response 1: We revised column heading of Tables and separated it with horizontal lines as recommended

2.     Table 1: Do you need a last column, “usage” since you indicated the use in your table title?

Response 2: We removed “Usage” from Table1.

3.     Table 3: Center the titles over the appropriate columns (Trachea, Kidney,  Cecal tonsils)

Response 3: We revised location of column

4.     Fig 1: X axis title indicates “hours” and, after each number it is not necessary to specify “h”

Response 4: We removed “h” in X-axis of Fig 1 as recommended

5.     Fig 1 Y axis: what do you mean by “Mean fold change of RNA copy/log2”?

Response 5: It means the log2 value of the average in 3 repeated experiments, but we changed “Mean fold change of RNA copy/log2” into “Log2 Fold change”

6.     Fig 1: You do not need a chart title since you have a figure legend with a title

Response 6: We deleted the chart title as recommended

7.     Fig 2: Label all the components of the reproductive tract in both IBV negative and IBV positive tracts. Consistently label the panels (IBV infected and uninfected), either on the top or bottom.

Response 7: We labeled all the components of the reproductive tract in both groups and changed the “DES+/KM91+” into “DES+/IBV+” as recommended 

Reviewer 2 Report

Most of the reviewers' comments have been considered and introduced in the revised manuscript version.

Author Response

Thank you..
